# Safe control under input limits with neural control barrier functions

**Simin Liu,  Changliu Liu,  John Dolan**
Robotics Institute
Carnegie Mellon University
`(siminliu, cliu6, jdolan)@andrew.cmu.edu`

**Abstract:** We propose new methods to synthesize control barrier function (CBF)-based safe controllers that avoid input saturation, which can cause safety violations. In particular, our method is created for high-dimensional, general nonlinear systems, for which such tools are scarce. We leverage techniques from machine learning, like neural networks and deep learning, to simplify this challenging problem in nonlinear control design. The method consists of a learner-critic architecture, in which the critic gives counterexamples of input saturation and the learner optimizes a neural CBF to eliminate those counterexamples. We provide empirical results on a 10D state, 4D input quadcopter-pendulum system. Our learned CBF avoids input saturation and maintains safety over nearly 100% of trials.

**Keywords:** safe control, input limits

## 1   Introduction

In theory, control barrier functions are an appealing tool for safe control. However, it is difficult to make the derived controllers respect input limits, which reduces their usage in practice. CBFs target the *set invariance* class of safety problems, in which safety is defined as keeping a system's state to a prescribed region. A large part of their appeal is that they offer mathematical guarantees of safety. Such assurances are essential for safety-critical robotics applications, like collision-free drone flight [1, 2], manipulators that work safely around humans [3], and stable bipedal walking [4]. However, these safety guarantees break down when input saturation occurs, since that means the system cannot exert the force required for an evasive maneuver. The system then becomes endangered (or dangerous), with the possibility of expensive equipment failure or people getting harmed. It is therefore imperative that we account for input limits when designing CBFs.

CBFs are *energy functions* which map states to an energy value, with safe states having lower energy. In *energy function methods*, an energy function is found and then a controller is crafted that dissipates the energy. The core problem of these methods is constructing the energy function. A valid energy function has to meet complex constraints that depend on the input limits, system dynamics, and safety specification. So far, this problem of designing CBFs around input limits has been studied to a limited degree, mostly for small or simple systems. To our knowledge, nothing has been proposed which handles the nonlinear and high-dimensional systems that are more realistic in robotics. Currently, many state-of-the-art approaches rely on hand-designing CBFs. This works well for certain simple systems, like the kinematic bicycle system [5, 6, 7, 8, 9, 10, 11, 12]. Some of these hand-design methods are more systematic, deriving non-saturating CBF for special *classes* of systems, like polynomial systems [13, 14]. Yet another variation of hand-design is to hand-select a parametric function for the CBF and optimize the parameters to avoid saturation [15, 16, 10]. The problem of designing a non-saturating CBF is also equivalent to computing a *control invariant set*, a well-known problem in the controls community [17]. This area has a long history and has been studied under different viewpoints and names, including viability kernel computation [18] and infinite-time reachable set computation [19]. For a condensed survey, see [20]. The most generic framework for computing control invariant sets is HJ Reachability [21]. Although it can handle nonlinear systems and gives formal guarantees against saturation, it cannot typically scale past systems

6th Conference on Robot Learning (CoRL 2022), Auckland, New Zealand.

of 6 or 7 dimensions in the state. This paper takes a different approach from HJ Reachability and abandons formal guarantees for better scalability.

For the most part, existing approaches for synthesizing non-saturating CBF apply to narrow classes of systems and can involve considerable human effort. In contrast, we envision a framework for *automating* CBF synthesis that has a *wide range of applicability*. To achieve this, we borrow ideas from machine learning (ML). We take inspiration from the related field of Lyapunov function (LF) synthesis, which has incorporated ML with success. LFs certify *stabilization*, rather than safety. However, finding a non-saturating CBF is essentially finding a function that satisfies a constraint on a set of inputs, which is the same problem as in LF synthesis. Recent works in LF synthesis represent the LF as a NN and then train it to satisfy the function constraints [22, 23, 24, 25]. We borrow this paradigm for the unique problem of input saturation of CBFs. For additional related work, please see Appendix Sec. 6.1.

There are several advantages to framing the problem as NN training. Firstly, it allows us to handle synthesis for nonlinear systems. Good non-saturating CBFs for nonlinear systems tend to be nonlinear functions, and NN are a richly expressive class of nonlinear functions. Secondly, it allows us to handle synthesis for systems with large state dimensions ($\geq 10D$). Neural networks can be trained quickly for inputs (here, the system state) of this size.

*We synthesize CBFs that respect input limits by posing this as a problem of training a neural function to satisfy limit-related constraints.* Our contributions are as follows:

1. A novel way to frame the synthesis of non-saturating CBF.

2. The design of a training framework, including a neural CBF design, loss function and training algorithm design.

3. Experimental validation on a 10D state, 4D input nonlinear system.

The rest of this paper is laid out as follows: Sec. 2 explains how CBFs work and carefully define the input saturation problem. Then, Sec. 3 details our approach, including the design of the neural CBF, training losses, and training algorithm. Finally, Sec. 4 describes how we test our method on a challenging quadcopter-pendulum system against several baselines.

## 2 Preliminaries

In this section, we provide a review of CBFs, mathematically define a non-saturating CBF, and explain the premise of CBF synthesis. First, some notation: for a function $c : \mathbb{R}^n \to \mathbb{R}$, let $\mathcal{C} \triangleq \{c\}_{\leq 0}$ be its zero-sublevel set, $\partial \mathcal{C} = \{c\}_{=0}$ the boundary of this set, and $\text{Int}(\mathcal{C}) = \{c\}_{<0}$ the interior. Now, we assume the following is given: (1) a control-affine system $\dot{x} = f(x) + g(x)u$, where $x \in \mathcal{D} \subset \mathbb{R}^n, u \in \mathcal{U} \subseteq \mathbb{R}^m$ and $f : \mathbb{R}^n \to \mathbb{R}^n$, $g : \mathbb{R}^n \to \mathbb{R}^{n \times m}$ are locally Lipschitz continuous on $\mathbb{R}^n$, (2) input set $\mathcal{U}$, a bounded convex polytope, and (3) a safety specification $\rho : \mathbb{R}^n \to \mathbb{R}$, which implicitly defines the *allowable set* as $\mathcal{A} \triangleq \{\rho\}_{\leq 0}$. Further, assume $\rho(x)$ is continuous and smooth. Given $\rho(x)$, we can define $r \in \mathbb{Z}^+$ as the *relative degree* from $\rho(x)$ to $u$ (i.e. the first derivative of $\rho(x)$ where $u$ appears).

We now walk through the process of producing a safe controller via CBF methods. In safe control, the goal is to keep some subset of the allowable set *forward invariant*, which means keeping any trajectory starting within the subset inside of it for all time. We call this subset the *safe set* and it will be defined by a function, the control barrier function, which we design. The CBF will also be used to define the safe controller that ensures forward invariance.

In the absence of input limits, we would just form a CBF as a known function of the safety specification $\rho(x)$:

$$\phi = \left[ \prod_{i=1}^{r-1} \left( 1 + c_i \frac{\partial}{\partial t} \right) \right] \rho \qquad \text{(limit-blind CBF)}$$

where $c_i < 0$. See [5, 26] for an explanation. The safe set $\mathcal{S} \subseteq \mathcal{A}$ defined by this limit-blind CBF is elaborated in Appendix Sec. 6.2. In the presence of input limits, we will have to modify this design, but we will come back to this. Next, to define a safe controller using a CBF is straightforward. A

safe controller simply needs to repel the system back into the safe set whenever it reaches the set boundary. The following theorem formalizes this idea:

**Theorem 1** (Taken from [5]). *Given a CBF $\phi$ and safe set $\mathcal{S}$, any feedback controller $k(x) : \mathbb{R}^n \to \mathbb{R}^m$ satisfying*

$$\dot{\phi}(x, k(x)) \triangleq \underbrace{\nabla\phi(x)^\top f(x)}_{L_f\phi(x)} + \underbrace{\nabla\phi(x)^\top g(x)}_{L_g\phi(x)} k(x) \leq 0 \quad \forall x \in \partial\mathcal{S} \tag{1}$$

*renders the system forward invariant over $\mathcal{S}$.*

Note that you can also require a CBF to satisfy a stricter inequality $\dot{\phi}(x) \leq -\alpha(\phi(x))$ for all $x \in \mathcal{D}$, where $\alpha(\cdot)$ is a class-$\kappa$ function. We elaborate on this alternative in Appendix Sec. 6.2. The theorem above requires the controller to repel the system (decrease $\phi$) from the boundary ($x \in \partial\mathcal{S}$). We are allowed to use any nominal controller, $k_{nom}$, as long as we modify its inputs to satisfy Eqn. 1 at the boundary. Thus, a CBF-based safe controller simply filters (modifies) a nominal controller online by applying the QP below at every step of control execution:

$$k(x) = \underset{u \in \mathcal{U}}{\arg\min} \ \frac{1}{2} \|u - k_{nom}(x)\|_2^2 \tag{CBF-QP}$$

$$\text{s.t.} \qquad L_f\phi(x) + L_g\phi(x)u \leq \begin{cases} 0 & \text{if } x \in \partial\mathcal{S} \\ \infty & \text{o.w.} \end{cases} \tag{2}$$

The issue with using a limit-blind CBF for this controller is that it can cause controller saturation. Specifically, saturation occurs when no $u \in \mathcal{U}$ exists that satisfies Eqn. 1 (constraint 2), causing the loss of safety guarantees.[1] What we need is to synthesize a *non-saturating CBF*, which is mathematically defined as:

**Definition 1** (*Non-saturating CBF*). *A function $\phi : \mathbb{R}^n \to \mathbb{R}$ is a non-saturating CBF over a set $\mathcal{S}$ if for all $x \in \partial\mathcal{S}$:*

$$\inf_{u \in \mathcal{U}} \dot{\phi}(x, u) \leq 0 \tag{3}$$

Intuitively, Def. 1 just requires that there exist a feasible control input to decrease $\phi$ (push the system to the interior of $\mathcal{S}$) at every state on the boundary, $\partial\mathcal{S}$. We approach this problem by modifying the limit-blind CBF:

$$\phi^* = \left[ \prod_{i=1}^{r-1} \left( 1 + c_i \frac{\partial}{\partial t} \right) \right] \rho - \rho + \rho^* \tag{modified CBF}$$

where $\rho^*(x) : \mathbb{R}^n \to \mathbb{R}$ such that $\{\rho^*(x)\}_{\leq 0} \subseteq \{\rho(x)\}_{\leq 0}$. This modification acts to shrink the associated safe set from $\mathcal{S}$ to $\mathcal{S}^*$. With a proper choice of function $\rho^*(x)$, we can exclude irrecoverable states (states where no feeasible input preserves safety) from the safe set boundary. In the rest of the paper, we focus on learning the function $\rho^*(x)$ to produce a non-saturating CBF.

**Problem scope:** to review, our proposed method applies to nonlinear, control-affine systems and safety problems that can be described by a smooth safety specification function $\rho(x)$. We also assume the system is deterministic and fully known. The method also applies to systems of high relative degree, since we based the modified CBF off of a higher-order CBF.

## 3 Learning Non-Saturating Control Barrier Functions

In this section, we present our neural CBF design (Sec. 3.1) and then discuss the training framework which optimizes it with respect to control limits (Sec. 3.2, 3.3). Learning is formulated as a min-max optimization problem of the following form, where $\theta$ denotes the parameters of $\phi^*$:

$$\min_{\theta} \max_{x \in \partial\mathcal{S}^*} \mathcal{L}(\theta, x) \tag{4}$$

---

[1]In practice, to avoid an unsolvable QP when saturation occurs, we add a slack variable to Eqn. 2. However, we will still violate safety guarantees.

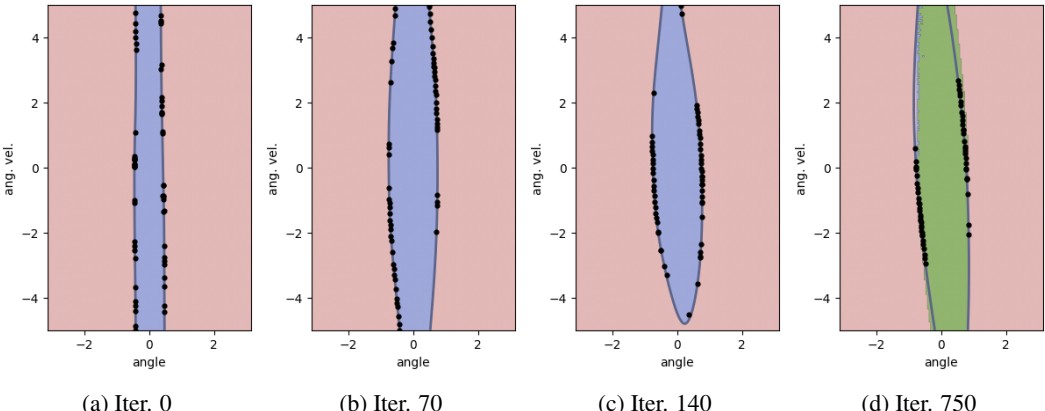

| (a) Iter. 0 | (b) Iter. 70 | (c) Iter. 140 | (d) Iter. 750 |

Figure 1: Learned safe sets for the toy cartpole problem at four iterations during training. Candidate counterexamples are marked in black. It is observed that the critic correctly identifies that the states with severe saturation are those with angle and angular velocity of the same sign (angular velocity swinging the pendulum out from the vertical). In (d), green denotes the *largest* non-saturating safe set, computed using MPC. Note that our volume enlargement is so effective that the learned safe set attains 95% of the largest volume.

We call this loss $\mathcal{L}(\theta, x)$ the *saturation risk*. This min-max problem is solved using a learner-critic algorithm (Alg. 1), where the critic and learner repeatedly find where the worst saturation occurs and then update the CBF to reduce saturation there. We also propose some strategies for training stably, boosting critic efficiency, and for increasing the volume of the safe set. For visualization, we plot the critic's counterexamples for a toy cartpole problem and intuitively justify their correctness (Fig. 1).

## 3.1 Neural CBF Design

Building upon the previous work discussed in Sec. 2, to design a non-saturating CBF, we only need to consider the design of the function $\rho^*(x)$ from the modified CBF. Let $\mathrm{nn} : \mathbb{R}^n \to \mathbb{R}$ be a multilayer perceptron with $\tanh$ activations. Then, we define

$$\rho^*(x) = (\mathrm{nn}(x) - \mathrm{nn}(x_e))^2 + \rho(x) \tag{5}$$

where $x_e$ is a state, identified by the user, that should belong to any reasonable learned safe set. Specifically, $x_e$ should belong to the allowable set $\mathcal{A}$ and should satisfy $\rho^{(i)}(x_e) \leq 0$ for $i \in [0, r-1]$ (this constrains any possible higher-order components in $x_e$; for example, velocities should be directed away from unsafe zones). For safety problems that limit the system's distance from an equilibrium, the equilibrium itself should lie in any reasonable safe set. For anti-collision problems, $x_e$ can be a point far from the ego robot. For additional recommendations, see the Appendix. We have designed $\rho^*$ to obey three constraints: *Constraint 1*: $\rho^*$ is smooth. This is required to preserve the smoothness of $\phi^*$, which allows us to make existence and uniqueness arguments for the closed-loop system. Our $\rho^*$ satisfies this because $\mathrm{nn}$ has smooth $\tanh$ activations and also $\rho$ is assumed smooth. *Constraint 2*: The 0-sublevel set of $\rho^*$ is contained within the 0-sublevel set of $\rho$: $\{\rho^*\}_{\leq 0} \subseteq \{\rho\}_{\leq 0}$. The allowable set can be shrunk but not enlarged; otherwise, dangerous states may be incorporated. Our design meets this criterion because $\rho^* \geq \rho$. *Constraint 3*: $\mathcal{S}^*$ is nonempty, where $\mathcal{S}^*$ is defined by $\rho^*$. With our design, $x_e \in \mathcal{S}^*$ by the definition of $\mathcal{S}^*$ (see Appendix) and our assumptions on $x_e$.

## 3.2 Training Framework

In the following sections, we present a loss function that encourages satisfaction of Eqn. 1 and the algorithm for solving the min-max problem on this loss. First, we define $\theta$ as the parameters of $\phi^*$, which include the weights of $\mathrm{nn}$ and the $c_i$ coefficients. Our loss function, called *saturation risk*, is defined as:

$$\mathcal{L}(\theta, x) \triangleq \inf_{u \in \mathcal{U}} \dot{\phi}^*_\theta(x, u) \tag{saturation risk}$$

---

**Algorithm 1** Learning non-saturating CBF

---

 1: **function** LEARN($\mathbb{X}_{ce}, \theta$)
 2:      Set learning rate $\alpha_l$
 3:      $\theta \leftarrow \theta - \alpha_l \cdot \nabla_\theta \left[ \sum_{x \in \mathbb{X}_{ce}} \text{softmax}(\mathcal{L}(\theta, x)) + \mathcal{R}(\theta) \right]$          ▷ From Eqn. 6, 7
 4:      **return** $\theta$
 5: **end function**
 6: **function** COMPUTECE($\theta$)
 7:      Set learning rate $\alpha_c$, number of gradient steps $N$
 8:      $\mathbb{X} \leftarrow$ uniformly sample a set on the boundary          ▷ See Alg. 2
 9:      **for** $i$ **in** $[0, \ldots, N]$ **do**
10:          $\mathbb{G} \leftarrow \nabla_\mathbb{X} \mathcal{L}(\theta, \mathbb{X})$          ▷ Batch gradient
11:          $\mathbb{P} \leftarrow$ project $\mathbb{G}$ along boundary
12:          $\mathbb{X} \leftarrow \mathbb{X} + \alpha_c \cdot \mathbb{P}$          ▷ Batch update
13:          $\mathbb{X} \leftarrow$ project $\mathbb{X}$ to boundary          ▷ See Alg. 3
14:      **end for**
15:      $\mathbb{X}_{ce} \leftarrow$ worst saturating states in $\mathbb{X}$
16:      **return** $\mathbb{X}_{ce}$
17: **end function**
18: **function** MAIN( )
19:      **Input:** dynamical system $\dot{x}$, safety specification $\rho$
20:      Randomly initialize neural CBF parameters $\theta$
21:      **Repeat:**
22:          $\mathbb{X}_{ce} \leftarrow$ COMPUTECE($\theta$)          ▷ $\mathbb{X}_{ce}$ is a set of counterexamples
23:          $\theta \leftarrow$ LEARN($\mathbb{X}_{ce}, \theta$)
24:      **Until** convergence
25:      **return** $\theta$
26: **end function**

---

It is a measure of the best-case saturation at a given state $x$. When $\mathcal{L}(\theta, x) \leq 0$, then no saturation occurs at $x$; when $\mathcal{L}(\theta, x) > 0$, it measures how severe the saturation is. Thus, our min-max problem (Eqn. 4) is to minimize the *worst best-case saturation* over the boundary. When the worst best-case is negative, i.e.

$$\max_{x \in \partial \mathcal{S}^*} \mathcal{L}(\theta, x) \leq 0 \qquad \text{(training goal)}$$

then we have successfully found a non-saturating CBF.

To solve the min-max problem on $\mathcal{L}(\theta, x)$ (Eqn. 4), we propose a learner-critic algorithm (Alg. 1). Essentially, the algorithm alternates between the critic computing counterexamples (maximization with respect to $x$) and the learner updating the CBF (minimization with respect to $\theta$). The critic uses projected gradient descent to produce an approximate maximizer, $\hat{x}^*$, and then the learner uses gradient descent to minimize the saturation loss at $\hat{x}^*$. Since both learner and critic perform gradient descent on $\mathcal{L}(\theta, x)$, it is useful to re-express it as an analytic function, rather than a continuous minimization. To find the analytic expression, observe that this $\mathcal{L}(\theta, x)$ is a minimization over $u$ where the objective is affine (from Eqn. 1) and the constraint set is a convex polyhedron $\mathcal{U}$, by assumption. This means the minimizing $u^*$ is one of the vertices $v \in \mathcal{V}(\mathcal{U})$ of the constraint set. Thus, $\mathcal{L}(\theta, x)$ can be computed as a discrete minimization, which is analytic:

$$\mathcal{L}(\theta, x) \triangleq \min_{v \in \mathcal{V}(\mathcal{U})} \dot{\phi}_\theta^*(x, v) \qquad \text{(analytic risk)}$$

### 3.3 Practical Training Methods

**Batch optimization:** in practice, training works better if both learner and critic use a *batch* of counterexamples, rather than just a single one. This means the critic optimizes a set of differently initialized counterexamples at once and the learner takes a weighted loss over a subset of the best counterexamples:

$$\theta = \theta - \alpha \cdot \nabla_\theta \left[ \sum_{x \in \mathbb{X}_{ce}} \text{softmax}(\mathcal{L}(\theta, x)) \right] \qquad (6)$$

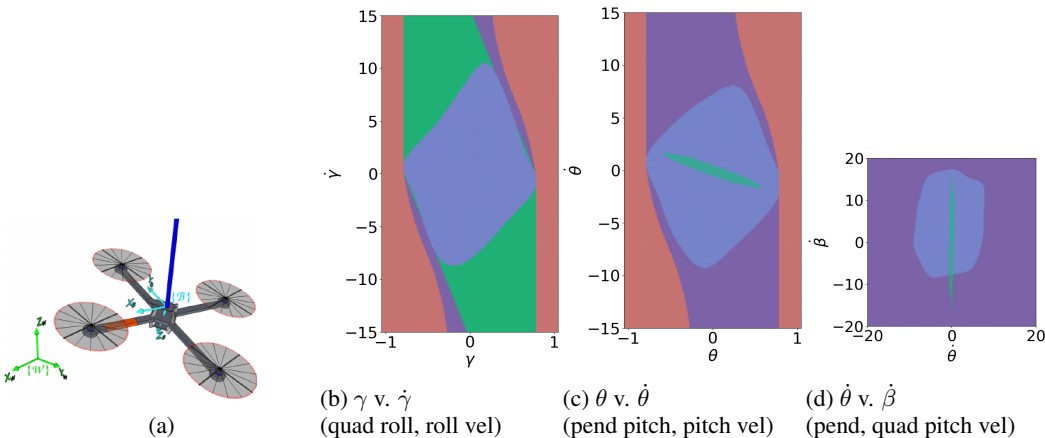

(a)

(b) $\gamma$ v. $\dot{\gamma}$
(quad roll, roll vel)

(c) $\theta$ v. $\dot{\theta}$
(pend pitch, pitch vel)

(d) $\dot{\theta}$ v. $\dot{\beta}$
(pend, quad pitch vel)

Figure 2: (Left) quadcopter-pendulum system (image from [30]). (Others) Axis-aligned 2D slices of the 10D safe set (blue is ours, purple is hand-designed CBF, green is safe MPC). For each slice, the unvisualized states have been set to 0.

This has the advantage of stabilizing convergence without adding much overhead. It stabilizes convergence by preventing (1) deadlock and (2) inaccurate gradients throughout training. Deadlock is when improvement at one counterexample causes saturation at another; averaging the learner's loss on a group of counterexamples avoids this by encouraging progress on many counterexamples at once. Inaccurate gradients refers to the fact that the learner should be using the gradient at the optimal counterexample $x^*$ ($\nabla_\theta \mathcal{L}(\theta, x^*)$), but since the critic is suboptimal, it uses a different gradient, $\nabla_\theta \mathcal{L}(\theta, \hat{x}^*)$. Clearly, this could derail training. However, we find that with batching, the critic produces better counterexamples, giving us a closer estimate of the true gradient ($\nabla_\theta \mathcal{L}(\theta, \hat{x}^*) \approx \nabla_\theta \mathcal{L}(\theta, x^*)$). Plus, with averaging, the learner averages out the effect of inaccurate gradients. We show in the Appendix that the algorithm requires large batches of counterexamples to perform well (Table 2). All in all, these techniques are a cheap and effective way to avoid the kind of training instability found in other counterexample-based methods, like adversarial training for image classifiers [27, 28, 29].

**Enlarging the safe set:** in this section, we introduce a regularization term that we add to the training objective to help enlarge the safe set. One measure of quality for safe sets is size. Since CBFs will allow a system to move freely inside a safe set, a larger safe set provides greater freedom towards accomplishing control objectives. Thus, a larger safe set enables better task performance. To this end, we add a regularization term to the training objective to encourage a large safe set. For context, recall that our CBF $\phi_\theta^*$ defines a safe set $\mathcal{S}^*$. Specifically, there is a function ss$^*$ of $\phi_\theta^*$ that implicitly defines $\mathcal{S}^*$ as its 0-sublevel set: $\mathcal{S}^* = \{x | \text{ss}^*(x) \le 0\}$; ss$^*(x)$ is defined in the Appendix. To clearly indicate its dependence on the CBF and its learned parameters $\theta$, we write it as ss$_\theta^*$ here. Next, we evaluate the regularization term at some sampled states $\mathbb{X}_{reg}$. We have:

$$\mathcal{R}(\theta) \triangleq \sum_{x \in \mathbb{X}_{reg}} \text{sigmoid}(\text{ss}_\theta^*(x)) \tag{7}$$

The idea behind the sigmoid is to encourage states near the boundary (ss$(\phi_\theta^*(x)) \approx 0$) to move (further) inside the safe set (ss$(\phi_\theta^*(x)) << 0$). Sigmoid gives a gradient which encourages values near zero to become (more) negative. We demonstrate in the Appendix that including this term can increase the volume by several factors.

## 4 Experiments

In this section, we train a neural CBF on a challenging nonlinear, high-dimensional, and input-limited robotic system. We pose two experimental questions:

**Q1.** How well do we achieve our training objective?

**Q2.** Does the CBF-based safe controller ensure FI?

| | Saturation at the boundary | | | Safety of rollouts w/ diff $k_{nom}$ | | | Safe set volume |
| --- | --- | --- | --- | --- | --- | --- | --- |
| | % non-sat. states | mean, std dev of sat. | worst sat. | $k_0$ | $k_{lqr}$ | $k_{lqr-agg}$ | (as fraction of ours) |
| **Ours** | 99.00 | $1.75 \pm 2.40$ | 15.19 | 99.62 | 99.62 | 99.02 | 1.00 |
| Hand-designed CBF | 78.68 | $4.99 \pm 3.78$ | 29.50 | 78.68 | 80.28 | 80.28 | 53.87 |
| Safe MPC | - | - | - | 99.06 | 99.54 | 99.44 | 0.08 |

Table 1: Comparison of our method against baselines. The "mean, std dev of sat." is $\mathbb{E}[\mathcal{L}(\theta, x)] \pm \sigma[\mathcal{L}(\theta, x)]$ and "worst sat." is $\mathcal{L}(\theta, x^*)$.

Our system is a pendulum on top of a quadcopter (Fig. 2). This is a coupled system, with the dynamics of both components found by the Euler-Lagrangian method [31]. The states are quadcopter position and roll-pitch-yaw orientation ($x, y, z$ and $\gamma, \alpha, \beta$) and roll-pitch pendulum orientation ($\phi_p, \theta_p$), as well as the first derivatives of these states. The inputs are thrust and torque ($F, \tau_\gamma, \tau_\beta, \tau_\alpha$), which are limited to a bounded convex polytope set. See the Appendix for the system dynamics. The safety specification is to prevent the pendulum from tipping and the quadcopter from rolling:

$$\rho = \gamma^2 + \beta^2 + \delta_p^2 - (\pi/4)^2 \tag{8}$$

where $\delta_p = \arccos(\cos(\phi_p) \cos(\theta_p))$ is the pendulum's angle from the vertical. Since the quadcopter position does not impact safety and the position dynamics can be decoupled, we exclude position and consider the resulting 10D state, 4D input system. Finally, in the design of $\rho^*(x)$, we let $x_e = \vec{0}$, which is the system's equilibrium.

Next, we propose metrics to answer each of our experimental questions:

**Q1 metrics.** *% of non-saturating states on $\partial \mathcal{S}^*$*: we measure how well we satisfy Eqn. 3 by uniformly sampling states on $\partial \mathcal{S}^*$ and calculating what percentage of them are non-saturating. We also use these samples to compute the *mean and variance of the severity of saturation*, $\mathbb{E}[\mathcal{L}(\theta, x)]$ and $\sigma[\mathcal{L}(\theta, x)]$. To approximate the *severity of the worst saturation*, $\mathcal{L}(\theta, x^*)$, we apply our critic and allow it to use more samples and computation time than during CBF training.

**Q2 metrics.** Q2 considers the in-the-loop control performance of our learned CBF. We measure *% of simulated rollouts that are FI* by initializing rollouts randomly inside $\mathcal{S}^*$ and simulating their trajectories under the safe controller (CBF-QP) until just after they reach the boundary. Then, we record whether the system exited or remained inside $\mathcal{S}^*$ after arriving at the boundary. The value of this metric depends on the choice of nominal controller, $k_{nom}$. We try $k_0(x) = 0$, which yields an unactuated system, and $k_{lqr}$ and $k_{lqr-agg}$, which are linear quadratic regulator (LQR) stabilizing controllers, with $k_{lqr-agg}$ tuned to be more aggressive.

We also choose two well-known alternatives as our baselines:

**Hand-designed CBF:** for CBFs, this is a typical alternative. We hand-pick a parametric CBF and then optimize the parameters for non-saturation.

**Safe MPC:** MPC is often used for safe planning and control and it can take input limits into account. The safety specification ($\rho \leq 0$) becomes a nonlinear state constraint and we also have to set the terminal constraint to be a smaller, known invariant set for the MPC solution to be forward invariant.

For details on any of these baselines, see the Appendix Sec 6.4. Note that safe MPC defines its safe set *implicitly*. This means the boundary of the safe set is not defined by a function. Hence, it would be too time-consuming to sample states on the boundary and compute the metrics for Q1. For safe MPC, we only report the results for Q2.

**Code.** Our code can be found at https://github.com/sliu2019/input_limit_cbf

**Training details.** The learning framework and metrics were implemented using Python and PyTorch [32]. Training took about 2 hours on a single NVIDIA GeForce RTX 2080 Ti GPU.

**Discussion.** As we can see in Table 1, for our learned CBF, almost 100% of the boundary states are non-saturating and almost 100% of the rollouts are FI across the nominal policies. This shows that our neural CBF and learning framework are an effective combination and capable of handling systems of high complexity. We are not able to attain 100% non-saturating states and FI rollouts, which is either due to suboptimality of training or limitations of the function class, as NNs are only universal approximators when their size is taken to the infinite. For our learned CBF, the severity of saturation at the saturating states is not negligible, but it is still low, and the worst saturation is large, but rarely encountered. The hand-designed CBF does poorly across all metrics (only 80%

of the boundary states are non-saturating and around $80\%$ of rollouts are FI). Safe MPC is equally good as the learned CBF at ensuring safety (almost 100% rollout safety)[2]. However, its significant disadvantage is that its safe sets are just a fraction of the size of our own ($8.4\%$ respectively). Size is an important measure of the quality of a safe set; these small sizes imply that this baseline can only ensure safety from relatively few states. The root of the issue is that safe MPC constructs a safe set by effectively expanding a smaller, "seed" invariant set provided by the user. The size of the safe set therefore depends greatly on the size of the seed invariant set. As we have already established, it can be quite difficult for users to design invariant sets (equivalently, non-saturating CBFs) of any reasonable size.

Visualizing slices of the safe set can provide deeper insight into the results of training (Fig. 2). Recall that a safe set contains only states that are *recoverable* from danger, given our input limits. We observe that our CBF has diamond-shaped safe sets in slices B and C, which makes sense because small angles can still be recoverable at larger speeds. In slice B, safe MPC's set largely captures states where the signs of $\gamma, \dot{\gamma}$ differ. This makes sense too, since these are states where the angular velocity acts to return the angle to 0. Safe MPC's set in slice B is also larger than ours. Since it is a non-saturating safe set, just like ours, we have to conclude that our algorithm could have found a larger non-saturating safe set. The reason for this suboptimality is that our volume regularization strategy is imperfect. It encourages expansion only at scattered points on the boundary, which means that some areas of the boundary may be unaffected. However, our regularization strategy seems to work well overall, since the volume of our safe set in 10D is much larger than that of safe MPC. In slices C and D, we get a glimpse of why that is. In both of these slices, safe MPC's set is tiny. It very tightly constraints the pendulum's angular velocity. While it makes sense for safe MPC to be more conservative towards the pendulum than the quadcopter (the pendulum is not directly actuated, while the quadcopter is), it is unnecessarily conservative.

Next, we analyze the shapes of the sets in slice D. Our safe set indicates that there should be a maximum safe angular speed for quadcopter and pendulum. This makes sense, since higher angular velocities are certainly harder to pull back from. We also observe a larger range for the quadcopter's pitch velocity than the pendulum's, which makes sense because again, the quadcopter is directly actuated and the pendulum is not. On the other hand, the hand-designed CBF does not restrict $\dot{\theta}$ or $\dot{\beta}$ in slice D. Due to the functional form of the hand-designed CBF (see Appendix Section 6.4), $\dot{\theta}$ and $\dot{\beta}$ are only restricted when $\dot{\theta}\theta > 0$ or $\dot{\beta}\beta > 0$ (that is, when angular velocity is strictly acting to destabilize). We've assumed $\theta, \beta = 0$ in slice D. This safety criterion is clearly too lenient, since large angular velocities that are currently swinging the system to the origin can cause overshooting and toppling shortly after. Overall, we can see that the non-neural CBF does not have the right function form. In general, it can be hard to guess what the right form might be for a system of this complexity. However, our algorithmic approach, combined with the NN functional representation, removes the need for guessing.

**Limitations and future work:** in the future, we intend to loosen our assumptions (see last paragraph of Sec. 2), particularly the assumption of a known and deterministic model. We would also like to test this method on more high-dimensional, nonlinear systems. One limitation of our work is that we had to perform a change of variables on the states input to the neural CBF before it would train successfully (see Appendix for details). While the dynamics on the new states were still nonlinear, this indicates that a simple feed-forward network might not be the best neural function class.

## 5   Conclusion

In summary, we proposed a framework for facilitating CBF synthesis under input limits. Thanks to our neural CBF representation and our effective and efficient learning framework, our method scales to higher-dimensional nonlinear systems. We learned a virtually non-saturating CBF on one such system, the quadcopter-pendulum. We hope that this safe control tool will makes CBFs more accessible and of practical value to roboticists.

---

[2]In theory, MPC should attain perfect safety. However, nonlinear MPC solvers are not "complete" in the sense that even if there exists a solution to their nonlinear program, they may not find it. Hence, sometimes they may falsely consider the safety problem infeasible and provide an unsafe input.

**Acknowledgments**

This material is supported by the United States Air Force and DARPA under Contract No. FA8750-18-C-0092 and the National Science Foundation under Grant No. 2144489. We thank Tianhao Wei, Weiye Zhao, Yiwei Lyu, and Qin Lin for useful conversations. We are also very grateful to Kate Shih, Qin Lin, Harrison Zheng, Raffaele Romagnoli, Tianhao Wei, Soumith Udatha, Tamas Molnar, and Jason Choi for reviewing drafts of this work.

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
