# OpenReview forum: "Safe Control Under Input Limits with Neural Control Barrier Functions"
_robot-learning.org/CoRL/2022/Conference — CoRL 2022 Poster_

### Official Review · Reviewer_A9jW · 2022-07-28

**Originality:** Good
**Technical Quality:** Good
**Clarity Of Presentation:** Fair
**Impact:** 3

**Recommendation:**

Weak Accept: I recommend accepting the paper, but will not argue for my recommendation if the majority of other reviewers have a different opinion.

**Summary:**

This paper presents a framework for learning the Input Constrained Control Barrier Function (CBF) for high-dimensional nonlinear systems. The authors formulate a min-max problem to learn the non-saturating CBF. The maximization problem focuses on generating worst-case data points, while the minimization problem attempts to learn the CBF. The authors validate their approach in simulations and demonstrate that the proposed method achieves a higher success rate ensuring safety.

**Issues:**

1. What is $x_e$ in line 116? Why author introduces Design 2 which is not used in the experiment?
2. In Figure 1, there is a noticeable difference between the learned set and the MPC set.
3. The explanation of the analytic saturation risk is not very clear. What is the difference between $\mathcal{V}(\mathcal{U})$ and $\mathcal{U}$?
4. “Another challenge to speed is systems with large state spaces…”
5. “We project the saturation risk’s gradient along the boundary, use this to update x, and finally project x back to the boundary.” How is the projection done? It would be beneficial to visualize the procedure to improve the understanding.
6. What is $s_{\theta}^*$, and how is it computed? What are $\mathbb{X}_{reg}$ and $\mathcal{D}$?

[1] Agrawal, Devansh R., and Dimitra Panagou. "Safe control synthesis via input constrained control barrier functions." *2021 60th IEEE Conference on Decision and Control (CDC)*. IEEE, 2021.

[2] Garg, Kunal, and Dimitra Panagou. "Control-lyapunov and control-barrier functions based quadratic program for spatio-temporal specifications." *2019 IEEE 58th Conference on Decision and Control (CDC)*. IEEE, 2019.

[3] Dalal, G., Dvijotham, K., Vecerik, M., Hester, T., Paduraru, C., & Tassa, Y. (2018). Safe exploration in continuous action spaces. *arXiv preprint arXiv:1801.08757*.

[4] Liu, P., Tateo, D., Ammar, H. B., & Peters, J. (2021). Robot reinforcement learning on the constraint manifold. In *Conference on Robot Learning* (pp. 1357-1366). PMLR.

[5] Akametalu, Anayo K., et al. "Reachability-based safe learning with Gaussian processes." *53rd IEEE Conference on Decision and Control*. IEEE, 2014.

[6] Bansal, Somil, et al. "Hamilton-jacobi reachability: A brief overview and recent advances." *2017 IEEE 56th Annual Conference on Decision and Control (CDC)*. IEEE, 2017.

**Quality Of The Limitations Section:**

Limitations are not well addressed

**Reviewer Expertise:**

4: The reviewer is confident but not absolutely certain that the evaluation is correct

**Robotics Focus:**

Highly relevant to robotics but no hardware experiments

**Strengths And Weaknesses:**

1. Insufficient baselines. There are several works focusing on the input saturation problem using CBF, e.g. [1, 2]. The authors only compare with a baseline that does not consider input constraints. This would be difficult to evaluate the actual contribution. More comparisons with proper baselines would be desirable.
2. The interpretation of the CBF design is vague. There are several questions that are not well discussed. Why was the Design 2 introduced and how is this design used? How does it perform in terms of training and ensuring safety?
3. Some limitations are not discussed. 1. how to handle systems with higher relative order? 2. how to tackle problems with multiple safety specifications? Input constraints can also be considered as safety specifications.
4. In the proposed approach, the generation of counterexamples is crucial. However, it is difficult to assess whether the proposed generation technique can be generalized to different tasks, e.g., where the boundary is non-convex. Perhaps the authors could try another task with more complex safety boundaries, such as a robot manipulation task.
5. Several sections are not clearly explained. Some notations are used without explanation.
6. Ablation studies are also beneficial to strengthen the paper.

**Summary Of Recommendation:**

Given the discussion in the strengths and weaknesses, I suggest that the authors should make a major revision. Improve the clarity of this paper, compare it with more appropriate baselines, and try different tasks to verify the generalizability of the proposed method.

---

> ### Author Response · Authors · 2022-08-27
> **Thanks for the thoughtful review (part 1)**
>
> **“Strengths and Weaknesses” section:**
>
> *“1. Insufficient baselines. There are several works focusing on the input saturation problem using CBF, e.g. [1, 2]. The authors only compare with a baseline that does not consider input constraints. This would be difficult to evaluate the actual contribution. More comparisons with proper baselines would be desirable.”*
>
> *One important clarification: our current baseline actually does consider input constraints.* It is a parametric CBF with parameters optimized to satisfy input constraints. It is actually a pretty reasonable baseline, since it is currently one of the only ways to design a CBF for a high-dimensional, nonlinear system.
>
> The reader suggests [1] as a baseline, which proposes hand-crafting a parametric CBF form and then setting the parameters via guessing or optimization. That is essentially the same as our existing baseline, but we will still include it shortly. *In general, the vast majority of works on synthesizing non-saturating CBFs are not eligible to serve as our baselines, since they don’t apply to nonlinear systems or are intractable for high-dimensional systems (see our introduction for a brief survey of such works).* This also applies to some works listed by the reviewer: the technique in [6] (Hamilton-Jacobi reachability) is known to be intractable for systems above 5D, and [5] relies on the same intractable technique in an RL setting (safe exploration).
>
> *The other works listed by the reviewer cannot be compared to ours and do not weaken our claim of novelty.* Specifically, [2] considers non-saturation for a different class of safety problems (prescribed-time set reachability). [3] is primarily a safe RL work that briefly suggests a naive way to account for input limits: shrink the safe set by a constant-width border, where the constant is guessed. [4] is another safe RL work which makes a significant simplifying assumption so it can treat input limits like state constraints. Perhaps this is why the reviewer suggests that input limits can be considered as state constraints in the general case (see our response below). This work assumes it is possible to directly set the velocity of multiple states (velocity inputs), which is rarely realistic for robotic systems. We do not make this assumption. However, we have found a few other eligible baselines, which we will include shortly. We have also cited the suggested works.
>
> *“2. The interpretation of the CBF design is vague. There are several questions that are not well discussed. Why was the Design 2 introduced and how is this design used? How does it perform in terms of training and ensuring safety?”
> And relatedly, from the “Issues” section: “What is $x_e$ in line 116? Why author introduces Design 2 which is not used in the experiment?”*
>
> The reviewer’s questions about the purpose of introducing an alternate CBF design, how this design is used, how it differs in performance, and what some notation in the design means are all answered in lines 117-129, where the following points are made:
>
> 1. The purpose: the different designs are meant for different subclasses of safety problems. Design 2 is for general problems, with the downside of having to enforce safe set non-emptiness by optimization (using the regularization term), rather than by construction. On the other hand, Design 1 is for special safety problems (like those which limit distance from an equilibrium), with the advantage of satisfying non-emptiness by construction.
> 2. How is Design 2 used? How does it differ in performance? Design 2 is optimized in the exact same way as the other design (same objective function and learner-critic algorithm). The choice between the two CBF designs doesn’t affect the efficacy of the learner-critic algorithm; the differences between the two designs are minor and the principles of the optimization algorithm are the same for both. To demonstrate this, we will apply Design 2 in the final draft.
> 3. What is $x_e$ in Design 1? As stated on lines 116 and 123-126, $x_e$ is a point, identified by the user, that should belong to any reasonable learned safe set. For example, for safety problems that limit distance from an equilibrium, we know that the equilibrium should belong to any reasonable safe set and we can let it be $x_e$.

---

> > ### Author Response · Authors · 2022-08-27
> > **Thanks for the thoughtful review (part 2)**
> >
> > **"Strengths and Weaknesses" section continued**
> >
> > *“3. Some limitations are not discussed. 1. how to handle systems with higher relative order? 2. how to tackle problems with multiple safety specifications? Input constraints can also be considered as safety specifications.”*
> >
> > 1. *An important clarification: our method handles systems of arbitrary relative order (see lines 77-80).* Our “limit-aware CBF” design (ln 102) incorporates the higher-order CBF design from [5]. Further, we demonstrated our method on a *degree 2* quadcopter-pendulum system.
> > 2. For simplicity’s sake, we only consider problems with a single safety specification for now. We’ve acknowledged this assumption in the appendix and will consider it further in future work. In the general case, input constraints can not be considered as additional safety specifications, which must be state constraints (see earlier related comment in response to “Strengths and Weaknesses” section).
> >
> > *“4. In the proposed approach, the generation of counterexamples is crucial. However, it is difficult to assess whether the proposed generation technique can be generalized to different tasks, e.g., where the boundary is non-convex. Perhaps the authors could try another task with more complex safety boundaries, such as a robot manipulation task.”*
> >
> > Actually, counterexample generation still works well for non-convex boundaries. In the intermediate stages of training, the boundary for the quadcopter-pendulum can become non-convex (see new Figure 4 in the appendix).  However, the counterexamples have sufficient quality to drive the boundary towards a good boundary, ultimately.
> >
> > This experimental observation confirms what should be true in theory: our counterexample generation does not rely on boundary convexity. This is because (1) we chose a boundary sampling technique which does not assume convexity and (2) applying gradient descent to the counterexamples along the boundary does not require convexity either.
> >
> > *“5. Several sections are not clearly explained. Some notations are used without explanation.
> > 6. Ablation studies are also beneficial to strengthen the paper.”*
> >
> > We are happy to clarify the paper further, but hopefully, the reader has had most questions addressed by our comments here. We have added an ablation study (see “Training details” section in the appendix).
> >
> > **“Summary of recommendations” section:**
> >
> > *“Improve the clarity of this paper, compare it with more appropriate baselines, and try different tasks to verify the generalizability of the proposed method.”*
> >
> > Absolutely! Hopefully we’ve addressed your clarity concerns here, and we’ll add baselines shortly. For the final draft, we are happy to devise another challenging, high-dimensional, nonlinear system, but we still think that our current experiments (cartpole and quadcopter-pendulum) are quite convincing.
> >
> > **"Issues" section:**
> >
> > 1. *“What is x_e in line 116? Why author introduces Design 2 which is not used in the experiment?”* See above.
> > 2. *“In Figure 1, there is a noticeable difference between the learned set and the MPC set.”* There is only a ~5% difference in volume, which is fairly trivial.
> > 3. *“The explanation of the analytic saturation risk is not very clear. What is the difference between $\mathcal{U}$ and $\mathcal{V}(\mathcal{U})$?”*  We defined $\mathcal{V}(\mathcal{U})$ in line 149. To summarize, $\mathcal{U}$ is the constraint set, assumed polyhedral, and $\mathcal{V}(\mathcal{U})$ is the set of vertices of the polyhedral constraint set.
> > 4. *“Another challenge to speed is systems with large state spaces…”* What is the issue?
> > 5. *‘“We project the saturation risk’s gradient along the boundary, use this to update x, and finally project x back to the boundary.” How is the projection done? It would be beneficial to visualize the procedure to improve the understanding.’* We defined the projection procedure in Algorithm 5 of the Appendix. Basically, it is difficult to project a point to an implicitly defined surface (that is, to find the point nearest on the surface to the original point). Thus, we approximately project (find a point near on the surface) using gradient descent. This works well, since our counterexamples never get very far from the surface during optimization. Great suggestion for visualizing the counterexample optimization process, we will include it in the final draft.
> > 6. *“What is $s_{\theta}^{\star}$, and how is it computed? What are $X_{reg}$ and $\mathcal{D}$?”* Thanks, we defined $s_{\theta}^{\star}$ in the appendix; it is simple to compute. $X_{reg}$ is defined in line 186 as a set of states sampled in the domain, $\mathcal{D}$ (defined line 72).

---

### Official Review · Reviewer_V3nm · 2022-08-01

**Originality:** Fair
**Technical Quality:** Good
**Clarity Of Presentation:** Good
**Impact:** 3

**Recommendation:**

Weak Reject: I recommend rejecting the paper, but will not argue for my recommendation if the majority of other reviewers have a different opinion.

**Summary:**

this paper proposes a method to construct Control Barrier Functions using Neural Networks (NN) to avoid input saturations. The keypoint is to ensure the condition satisfaction on the boundary of the safe set, hence they formulate a min-max optimization problem with each step some counter-examples generated from the sampling process. As for the experiment, the authors design a 10d quadrotor-pendulum example and achieve better “satisfaction rate” and “forward invariant rollout rate” than non-NN CBF baseline approach.


**Issues:**

1. Weak baseline: I doubt the strength of this baseline. There are lots of related works [1-4] in using Neural Networks to construct CBF and have witnessed great success. Since NN is not a novelty of this paper, it is unfair to only compare a non-NN CBF. Besides, I would also suggest comparing methods such as reinforcement learning.
2. Lack of experiments: The system used in the paper has an equilibrium point so the CBF is constructed following design-1. But the paper also introduces the design-2 which is not implemented in the experiment. More experiments are expected to also support that design choice.
3. Lack of ablation studies: We would like to see how each of the components / tricks in the paper leads to the performance gain.

[1] Wang, Chuanzheng, et al. "Learning control barrier functions with high relative degree for safety-critical control." 2021 European Control Conference (ECC). IEEE, 2021.
[2] Qin, Zengyi, et al. "Learning safe multi-agent control with decentralized neural barrier certificates." arXiv preprint arXiv:2101.05436 (2021).
[3] Meng, Yue, Zengyi Qin, and Chuchu Fan. "Reactive and safe road user simulations using neural barrier certificates." 2021 IEEE/RSJ International Conference on Intelligent Robots and Systems (IROS). IEEE, 2021.
[4] Xiao, Wei, et al. "Barriernet: A safety-guaranteed layer for neural networks." arXiv preprint arXiv:2111.11277 (2021).



**Quality Of The Limitations Section:**

Additional details required

**Reviewer Expertise:**

5: The reviewer is absolutely certain that the evaluation is correct and very familiar with the relevant literature

**Robotics Focus:**

Sufficient demonstration on hardware

**Strengths And Weaknesses:**

Strengths:
1. The paper is easy to read - it provides detailed thoughts on how to guide the training more efficiently and more stably.
2. Unlike papers working on only low-dimension systems, they work on a challenging 10d quadrotor-pendulum dynamic.

Weaknesses:
1. The approach is only testified on one experiment
2. The approach is only compared with one baseline
3. No hardware experiments or ablation studies (for different design choices and improvements)


**Summary Of Recommendation:**

I recommend rejecting this paper. Although the authors provide an interesting algorithm to construct CBF for input saturations, they lack thorough and solid experiments to persuade the readers about this approach’s benefit. The baseline is weak (only a CBF constructed in a non-NN fashion) and the experiment is only considering the case where an equilibrium point is available. The experiment cannot fully convince us why we need this new approach.

---

> ### Author Response · Authors · 2022-08-27
> **Thanks for the thoughtful review (part 1)**
>
> **“Strengths and weaknesses” section:**
>
> *“1. The approach is only tested on one experiment
> 2. The approach is only compared with one baseline
> 3. No hardware experiments or ablation studies”*
>
> For the final draft, we are happy to devise another challenging, high-dimensional, nonlinear system, but we still think that our current experiments (cartpole and quadcopter-pendulum) are quite convincing. We will add baselines shortly; we have added ablation studies (see “Training details” section in the appendix).
>
> **“Summary of recommendation” section:**
>
> *“Although the authors provide an interesting algorithm to construct CBF for input saturations, they lack thorough and solid experiments to persuade the readers about this approach’s benefit. The baseline is weak (only a CBF constructed in a non-NN fashion) and the experiment is only considering the case where an equilibrium point is available. The experiment cannot fully convince us why we need this new approach.”*
>
> Our current baseline is actually pretty reasonable (see below), but we will add more baselines shortly.
>
> **“Issues” section:**
>
> *“1. Weak baseline: I doubt the strength of this baseline. There are lots of related works [1-4] in using Neural Networks to construct CBF and have witnessed great success. Since NN is not a novelty of this paper, it is unfair to only compare a non-NN CBF. Besides, I would also suggest comparing methods such as reinforcement learning.”*
>
> Our baseline is actually pretty reasonable. It is a parametric CBF with parameters optimized to satisfy input limits. It is currently one of the only ways to design a CBF for a high-dimensional, nonlinear system.
>
> Also, we did not claim to invent neural CBFs (ln 49-51), which are ubiquitous lately, especially in reinforcement learning (RL). Instead, our core insight is to apply neural CBFs to an open problem in the robotics control field, which is:
> 1. Synthesizing *non-saturating* CBF for *high-dimensional, nonlinear* systems, which are much harder to handle efficiently than small and simple toy systems
> 2. Note: we are specifically interested in synthesizing *CBFs*, not safe policies. CBFs are a *far more generic tool* than a safe policy. They produce a “safety layer” that can be placed on top of *any arbitrary* policy, modifying its inputs to convey mathematical guarantees of safety. If the control task changes (like from goal reaching to trajectory following), our CBF can still be used, but a safe policy would have to be relearned in this case.
>
> By reframing the classical controls problem of constructing non-saturating CBF as a deep learning problem, we were able to solve it at much higher dimensions and for more general kinds of systems than ever before.
>
> *We are the first to apply neural CBFs to this problem, to the best of our knowledge.* Other works to apply neural CBFs, including the ones the reviewer listed, have applied them to problems unrelated to our own. For example, [R3] learns an unknown safety criterion from safe expert trajectories, [1] indirectly learns unknown system dynamics by learning CBF dynamics, [2] jointly learns a safe policy and its safety certificate for multiagent reinforcement learning (RL), [3] applies this same joint learning technique to simulating safe drivers, [4] learns how to reduce the safety intervention to improve performance at non-safety objectives, like tracking. In fact, most of the works that apply neural CBFs assume control limits don’t exist (like [1, 3, 4]) and [2] only briefly suggests a heuristic for producing a safe input within limits. *Therefore, it is not ideal to compare against other neural CBF works, since they don’t account for input limits.*
>
> *The reviewer also suggests an RL baseline, which is not appropriate, as we are not learning a safe policy.* As mentioned earlier, CBFs and safe policies are completely different tools. CBFs convey guaranteed safety to any policy, which is much more powerful than having a particular safe policy. In our paper, we showed that our learned CBF ensures basically the same level of safety across different nominal policies (Table 1). *Thus, comparing to an RL baseline would be an “apples to bananas” comparison.* However, we will provide other baselines shortly and we have cited the safe RL works.
>
> [R3] Robey, Alexander, et al. "Learning control barrier functions from expert demonstrations." 2020 59th IEEE Conference on Decision and Control (CDC). IEEE, 2020.
>
> [R4] Lindemann, Lars, et al. "Learning hybrid control barrier functions from data." arXiv preprint arXiv:2011.04112 (2020).

---

> > ### Author Response · Authors · 2022-08-27
> > **Thanks for the thoughtful review (part 2)**
> >
> > **“Issues” section continued:**
> >
> > *“2. Lack of experiments: The system used in the paper has an equilibrium point so the CBF is constructed following design-1. But the paper also introduces the design-2 which is not implemented in the experiment. More experiments are expected to also support that design choice.
> > 3. Lack of ablation studies: We would like to see how each of the components / tricks in the paper leads to the performance gain.”*
> >
> > Regarding Design 2, please see our related comments to Reviewer A9jW.  We have added ablation studies (see “Training details” section in the appendix).

---

### Official Review · Reviewer_wTA7 · 2022-08-01

**Originality:** Very Good
**Technical Quality:** Excellent
**Clarity Of Presentation:** Very Good
**Impact:** 4

**Recommendation:**

Weak Accept: I recommend accepting the paper, but will not argue for my recommendation if the majority of other reviewers have a different opinion.

**Summary:**

This paper proposes a modification to the Control barrier function (CBF) formulation to allow for non-saturating controls, i.e. controls that do not exceed the actuation limits of the controller. The authors go about this by first characterizing a non-saturating invariant set, and then propose a learning method to learn the parameters of a control barrier function that will ensure a non-empty safe set and a resulting control law to keep the system within the safe set limits. Note that the authors assume that the dynamics of the system are known, and therefore this work is more in line with optimization based control rather than learning about the system dynamics.

**Issues:**

Please refer to the main review.

**Quality Of The Limitations Section:**

Limitations are addressed clearly

**Reviewer Expertise:**

3: The reviewer is fairly confident that the evaluation is correct

**Robotics Focus:**

Highly relevant to robotics but no hardware experiments

**Strengths And Weaknesses:**

**Strengths:**
* Strong theoretical derivation and proofs ensure soundness of the work. While I am not aware of the full background of control barrier functions, I have checked the derivations and proofs within the paper. Prior knowledge was well referenced in the paper.
* Useful research direction: Controller saturation, and related simplistic modeling of actuation systems is a key limitation for a lot of simulation based robotics research and it is nice to see work addressing the issue.
* Challenging kinodynamic environment.

**Drawbacks:**
* Assumptions of knowledge of the system dynamics: As with any planning based approach, the requirement of knowing the system dynamics requires intensive efforts in system identification, and even then often the dynamics are only known with uncertainty. How senstive would the system be to misidentification of the dynamics? Perhaps the authors might be able to include an experiment on that.
* Affine control law: How frequently is the assumption of an affine control violated in practise? What are the typical ways in which this is addressed?

Comments:
* The authors seem to have adopted an approach where the system starts with a saturating safe-set and works inwards from there. Would the authors comment on a reversed approach where they grow the safe set from a smaller starting safe set? I raise this question as even though the authors present results where the learned FI set was almost entirely safe, there were still saturated control inputs.


**Summary Of Recommendation:**

I found the presentation of the paper to be quite clear, the problem is well motivated and the research question is important. The prior works have been referenced, and the novelty of the paper is well established in the writing. I would recommend an accept, but as a non-expert in the field.

---

> ### Author Response · Authors · 2022-08-27
> **Thanks for the thoughtful review**
>
> **“Strengths and weaknesses” section:**
>
> *“Drawback 1: Assumptions of knowledge of the system dynamics: As with any planning based approach, the requirement of knowing the system dynamics requires intensive efforts in system identification, and even then often the dynamics are only known with uncertainty. How sensitive would the system be to misidentification of the dynamics? Perhaps the authors might be able to include an experiment on that.”*
>
> See response to reviewer 8LBY.
>
> *“Drawback 2: Affine control law: How frequently is the assumption of an affine control violated in practise? What are the typical ways in which this is addressed?”*
>
> Here, like in the majority of CBF works, we assume a control-affine system (that is, a system of form $\dot{x} = f(x) + g(x) u$). It’s an unrestrictive assumption, as non-affine dynamics can always be converted to affine form by just redefining the system.
>
> *“Comments: The authors seem to have adopted an approach where the system starts with a saturating safe-set and works inwards from there. Would the authors comment on a reversed approach where they grow the safe set from a smaller starting safe set? I raise this question as even though the authors present results where the learned FI set was almost entirely safe, there were still saturated control inputs.”*
>
> That’s a very interesting point! There are some works along these lines in the controls literature (they don’t involve learning) [R1, R2, R3]. Their basic idea is that given a small, known invariant set (safe set), we can expand it to include all the states within short reach of it. (The small invariant set can be produced in various ways: perhaps we have a naive safety controller, with an invariant set which can be defined analytically or computed.) These approaches are not iterative per se, but it could be possible to reimagine this as an iterative process, where the invariant set is expanded again and again.
>
> [R1] Chen, Yuxiao, et al. "Backup control barrier functions: Formulation and comparative study." 2021 60th IEEE Conference on Decision and Control (CDC). IEEE, 2021.
>
> [R2] Gurriet, Thomas, et al. "An online approach to active set invariance." 2018 IEEE Conference on Decision and Control (CDC). IEEE, 2018.
>
> [R3] González, Alejandro H., and Darci Odloak. "Enlarging the domain of attraction of stable MPC controllers, maintaining the output performance." Automatica 45.4 (2009): 1080-1085.

---

### Official Review · Reviewer_8LBY · 2022-08-08

**Originality:** Very Good
**Technical Quality:** Excellent
**Clarity Of Presentation:** Very Good
**Impact:** 4

**Recommendation:**

Weak Accept: I recommend accepting the paper, but will not argue for my recommendation if the majority of other reviewers have a different opinion.

**Summary:**

This paper proposes a method for synthesizing control barrier functions (CBF) that avoids action saturation for safe controllers. The main contribution is the proposal of a limit-aware CBF, and the usage of a min-max learning framework that uses a critic to find worst-case saturation points and a learner for updating the CBF. They also propose two neural CBF designs, an efficient method for optimizing the learner as a discrete minimizing problem, and introduce a regularization term to encourage larger safety sets. The paper compares their method to a non-neural CBF baseline for a pendulum-balancing task on a quadcopter, and found that their method enables nearly 100% of the boundary states of the safe set to avoid input saturation and nearly 100% of the rollouts are forward invariant inside the safe set, which is a significant improvement over the baseline.

**Issues:**

* Additional experiments that include stochasticity to the dynamics of the simulated experiments would be more convincing for understanding whether the existing technique is potentially applicable to robot hardware.
* A discussion on how challenging and important hyperparameter optimization is in this method for learning (e.g: including the regularization term, how many samples are warmstarted vs. taking from best of a batch, etc.)

Minor issues include:
* Replacing related works that are cited as being on arxiv with the peer reviewed versions (e.g: Clark, Andrew. "Verification and synthesis of control barrier functions." 2021 60th IEEE Conference on Decision and Control (CDC). IEEE, 2021.)
* The x and y axis titles on Figure 2 (b), © and (d) are quite small and enlarging them would improve readability


**Quality Of The Limitations Section:**

Limitations are addressed clearly

**Reviewer Expertise:**

2: The reviewer is willing to defend the evaluation, but it is quite likely that the reviewer did not understand central parts of the paper

**Robotics Focus:**

Highly relevant to robotics but no hardware experiments

**Strengths And Weaknesses:**

Strengths
* The introduction of a non-saturating CBF to avoid input-saturation for safe controllers, which is an important consideration when synthesizing CBFs.
* The two proposed neural CBF designs are mathematically well motivated by the three constraints they need to satisfy to be valid safety specifications.
* The two strategies of taking the best of a batch and warmstarting the optimizing process for finding the states along the safe set boundary where saturation occurs is a useful inclusion to the process for improving the critic.
* The re-expression of the saturation risk as an analytical function and framing the learner update rule as a discrete minimization problem over counterexamples is both efficient and well-thought out.
* The regularization term that encourages states near the safe set boundary to move inside the safe set (or further in) is clever and well-designed in this paper.

Weaknesses:
* This paper does not implement their method on robot hardware, which makes it hard to determine how effective it is in practice due to the limiting assumptions on known and deterministic dynamics. In lieu of this, it would be worthwhile and more convincing to at least see simulated experiments with noise added to the dynamics to see how the learned neural CBF performs.


**Summary Of Recommendation:**

This paper addresses the important problem of avoiding input saturation when synthesizing control barrier functions (CBF) for safety specifications. It introduces a novel CBF formulation and a learning framework for learning to identify worst-case saturation states at the boundary of the safety set and improving the CBF. In addition, the paper includes many additional components to improve the efficiency and performance of the learning method, and demonstrates the efficacy of the proposed solution on a challenging pendulum-balancing domain with a simulated quadcopter. The paper does a sound job of mathematically motivating the design choices in this paper, and for these reasons I suggest this paper be accepted.

---

> ### Author Response · Authors · 2022-08-27
> **Thanks for the thoughtful review**
>
> **“Strengths and weaknesses” section:**
>
> *“This paper does not implement their method on robot hardware, which makes it hard to determine how effective it is in practice due to the limiting assumptions on known and deterministic dynamics. In lieu of this, it would be worthwhile and more convincing to at least see simulated experiments with noise added to the dynamics to see how the learned neural CBF performs.”*
>
> Yes! Our ultimate plan is to transfer this work to a real-life robot. In this paper, since the problem of constructing a non-saturating CBF for high-dimensional, nonlinear systems is already quite challenging, we made some simplifying assumptions. This is mainly the assumption about known and deterministic dynamics. In general, our goal is to gradually loosen those assumptions in follow-up projects. For this project, we have added additional experiments on robustness of the learned CBF to stochastic and mismodeled dynamics (see “Testing details” section in the appendix).
>
> **“Issues” section:**
>
> *“1) Additional experiments that include stochasticity to the dynamics of the simulated experiments would be more convincing for understanding whether the existing technique is potentially applicable to robot hardware.”*
>
> As mentioned above, we added additional experiments on robustness of the learned CBF to stochastic and mismodeled dynamics. Since the learned CBF is not particularly robust, as expected, we also give an idea for how we might account for this in the future.
>
> We consider dynamics with state-dependent disturbance $d(x) \in \mathcal{D}$, where $\mathcal{D}$ is a compact set:
> $\dot{x} = f(x) + g(x)u + d(x)$
> This disturbance can generically capture model mismatch, dynamics noise, or actual external disturbances. Then, we formulate the new min-max optimization as:
> $\min_{\theta} \max_{x \in \partial \mathcal{S}^{*}} \mathcal{L}(\theta, x, d(x))$
> Intuitively, this means finding counterexamples of saturation under worst-case disturbances (mismatch, noise). We would need to redesign the critic to solve this new inner maximization problem.
>
> *“2) A discussion on how challenging and important hyperparameter optimization is in this method for learning (e.g: including the regularization term, how many samples are warmstarted vs. taking from best of a batch, etc.)”*
>
> We have included an ablation study on a few key design choices (see “Training details” section in the appendix). Hyperparameter optimization is important, but fairly straightforward. In our ablation study, the majority of hyperparameter choices yielded successful training (the objective basically reaches minimum value, 0). Thus, tuning the hyperparameters is just for performance (i.e. speeding up training, increasing the safe set volume).
>
> *“Minor issues include:
> Replacing related works that are cited as being on arxiv with the peer reviewed versions (e.g: Clark, Andrew. "Verification and synthesis of control barrier functions." 2021 60th IEEE Conference on Decision and Control (CDC). IEEE, 2021.)
> The x and y axis titles on Figure 2 (b), © and (d) are quite small and enlarging them would improve readability”*
>
> Done, thanks.

---

### Author Response · Authors · 2022-08-27
**Paper with updates**

We have made the reviewers' requested changes and added a number of the requested experiments (more to come).

---

> ### Author Response · Authors · 2022-08-28
> **More updates**
>
> We were not able to complete running the baselines.
> The planned baselines were:
> 1. **Nonlinear MPC**: this is a non-CBF method, but it can theoretically be used to a similar effect (acting like a “safety layer” rather than a safe policy, see comment to reviewer V3nm). In general, nonlinear MPC can be used, with difficulty, for input-limit-aware safe controls. We can formulate the objective function for safety and add input constraints, which can be handled natively by an MPC solver. However, the caveat is that it will not ensure forward invariance unless we impose a special terminal constraint on the states (namely, that the final state lands in a known invariant set). This poses great difficulty, as finding such an invariant set is actually precisely the problem we’ve set out to solve in our paper - it is rather circular. One way to “cheat” this problem for quadcopter-pendulum is to let the terminal constraint be the equilibrium point. While this ensured nearly 100% safety under limits on a limited number of rollouts, the resulting safe set was very conservative (about 1/200 the size of ours). This illustrates another advantage of our method: we can ensure nearly 100% safety under input limits over a much larger state set than other methods, thanks to our regularization term.
> 2. **[1] from reviewer A9jW**. This baseline is quite similar to our own: both involve hand-crafting a parametric CBF form and then setting the parameters via guess-and-check in the case of [1] and optimization in the case of our baseline. We did not successfully run this baseline, as we were having trouble hand-designing a CBF with a non-empty safe set. (CBFs with empty safe sets are obviously invalid). [1] admits that the example CBFs were somewhat arbitrary in design and provides little guidance. We tried upwards of 30 designs, to no avail. This actually illustrates the weakness of CBF synthesis approaches requiring hand-design: it can be very suboptimal and unintuitive, even for someone with a controls background.
>
> Due to short turnaround time (2 weeks) and our other rebuttal tasks, we could not totally complete the baselines.
> We believe that our current results (on cartpole, quadcopter-pendulum) are quite strong on their own; indeed, the nearly 100% safety on quadcopter-pendulum would be hard to beat.
> However, we would be happy to wrap up the baselines for the final draft.

---

### Meta-Review · Area_Chair_1a4q · 2022-08-08

**Recommendation:** Accept (Poster)
**Confidence:** 2

**Metareview:**

The authors propose an interesting work building on the control barrier functions.
The methods seem to be sound and the experiment in a high-dimensional setting, the quadrotor one, seems to be very convincing.
The work requires the knowledge of system dynamics: while this is a limitation, it is a common assumption of a robotics system and doesn't invalidate the importance of the proposed work. The reviewers ask for a bit of discussion on the topic, in particular focusing on the possibility of an inaccurate model and how this will impact the learning system.

The main issues of this paper are the lack of citations of recent methods in safe reinforcement learning and the lack of baselines in the experimental evaluation.

For the acceptance of this paper, the authors must update their paper with all the suggested citations from the reviewers and add additional baselines to the experimental evaluation.

Reviewers suggest also adding some more ablation studies to show the robustness of the approach. This would strengthen the quality of the work.

A minor issue of this work is the lack of clarity in some notation. Authors should make sure that the notation is introduced properly.

========================================================================================================

The reviewers agree on the fact that the paper is interesting and positively value the novelty of the approach. Most reviewers lean towards acceptance of the paper. Unfortunately, one reviewer is not convinced about the strength of the current experimental evaluation, in particular the lack of appropriate baselines.

While I think that the reviewer is right in highlighting this drawback, I think the authors should be able to implement the baseline for the final version of the paper. Indeed, the proposed task is quite complex, thus making it difficult to implement such a baseline in the restricted timeframe of the rebuttal.
Furthermore, the authors put reasonable effort into the rebuttal phase.

This paper is clearly borderline, but it may be worth considering accepting it, at least for a poster presentation, if the authors are able to complete the required baselines for the camera-ready version.

A minor, but important, note: in case this paper is accepted, the authors must fix the formatting. In the latest version, the appendix appears before the bibliography, but this should not happen in the camera-ready.

**Best Paper Nomination:**

No

---

> ### Author Response · Authors · 2022-08-27
> **Thanks for the thoughtful metareview**
>
> *“The reviewers ask for a bit of discussion on the topic, in particular focusing on the possibility of an inaccurate model and how this will impact the learning system.”*
>
> In the responses below, we discussed how we might eliminate these assumptions in subsequent work. We have also included experiments involving stochastic and mismodeled dynamics (see “Testing details” section in the appendix).
>
> *“The main issues of this paper are the lack of citations of recent methods in safe reinforcement learning and the lack of baselines in the experimental evaluation. For the acceptance of this paper, the authors must update their paper with all the suggested citations from the reviewers and add additional baselines to the experimental evaluation.”*
>
> We have added the safe RL citations from reviewers A9jW and V3nm and will add a few extra baselines shortly.
>
> *“Reviewers suggest also adding some more ablation studies to show the robustness of the approach. This would strengthen the quality of the work. A minor issue of this work is the lack of clarity in some notation. Authors should make sure that the notation is introduced properly.”*
>
> We added some ablation studies (see “Training details” section in the appendix) and defined a few notations more clearly.